

# Impact of Satellite-Based Ice Surface Temperature Initialization on Arctic Winter Forecasts Using the Korean Integrated Model

Eui-Jong Kang[1], Byung-Ju Sohn[1,2], Wonho Kim[3], Young-Chan Noh[4], Sihye Lee[3], In-Hyuk Kwon[3], Hwan-Jin Song[5]

[1]School of Earth and Environmental Sciences, Seoul National University (SNU), Seoul, South Korea
[2]School of Atmospheric Physics, Nanjing University of Information Science and Technology (NUIST), Nanjing, China
[3]Korea Institute of Atmospheric Prediction Systems (KIAPS), Seoul, South Korea
[4]Korea Polar Research Institute (KOPRI), Incheon, South Korea
[5]Department of Atmospheric Sciences, Kyungpook National University (KNU), Daegu, South Korea

*Correspondence to*: Eui-Jong Kang (kangej@snu.ac.kr)

**Abstract.** Ice surface temperature (IST) is critical for representing surface energy exchange in Arctic forecasts, yet its initialization in operational numerical weather prediction (NWP) systems remains overly simplified—often inherited from background states or prescribed as spatially uniform values—due to the scarcity of reliable, spatiotemporally continuous observations. This study examines the forecast impact of realistic IST initialization using the Korean Integrated Model (KIM),

a global NWP system that performs well at mid- and low-latitudes but shows limited forecast skill over the Arctic, particularly during winter. In its operational configuration, KIM uniformly initializes a fixed IST value of 271.35 K (−1.8 °C) over all sea ice-covered regions, making it a suitable testbed for this investigation. We generate a physically consistent, gap-free IST dataset using a standalone sea ice model nudged with satellite-retrieved ISTs and initialize it into KIM. Numerical experiments for the 2021–2022 Arctic winter show that the control run, despite being initialized with unrealistically warm IST, exhibits a

slight cold bias in the lower troposphere—indicating an inherent cooling tendency in KIM that counterbalances artificial heating from the surface boundary. The experimental run, initialized with realistic IST, further amplifies this cold bias. Although random errors are reduced by 3–5% in the lower atmosphere, the intensified bias ultimately degrades overall forecast performance. These results demonstrate that while realistic IST initialization influences short-range Arctic forecasts, its benefits are limited without concurrent improvements in underlying model bias. The findings underscore the need for parallel

improvements in internal model processes to fully realize its benefits, thereby offering guidance for achieving meaningful gains in Arctic forecast accuracy within KIM.

## 1 Introduction

With the acceleration of global warming, the Arctic—one of the most climatically sensitive regions—has undergone rapid and unprecedented changes in recent decades (IPCC, 2013; AMAP, 2021). These changes have heightened the demand for accurate

short- to medium-range weather forecasts in the polar regions, not only for safeguarding local communities and infrastructure and supporting maritime navigation and sustainable resource development in the Arctic Ocean (Pregnolato et al., 2016; Curtis



et al., 2017; Constable et al., 2022), but also for better understanding potential linkages to mid-latitude weather extremes (Francis and Vavrus, 2012; Screen and Simmonds, 2013; Cohen et al., 2014; Jung et al., 2014). However, forecasting skill in the Arctic remains relatively limited due to a combination of limited factors, including sparse conventional observations,

challenges in representing polar-specific processes, and simplified coupling among atmosphere, ocean, and sea ice components (Jung et al., 2016; Ortega et al., 2022). As a result, numerical weather prediction (NWP) models tend to exhibit larger forecast errors in the Arctic than in mid- and low-latitude regions (Jung et al., 2016; Jung and Matsueda, 2016)

Among various contributing factors, the initialization of surface boundary condition—particularly ice surface (or skin) temperature (IST)—has received relatively little attention despite its important role in the Arctic surface energy budget. IST

directly controls the surface energy balance and strongly influence turbulent heat fluxes, surface-based temperature inversions, boundary-layer stability, and low-level cloud formation (Lesins et al., 2012; Kral et al., 2021; Arduini et al., 2022; Sledd et al., 2025). Nevertheless, IST initialization in operational models remains overly simplistic, often inherited from background states or initialized with spatially uniform values. This simplification largely stems from the scarcity of reliable, continuous observations.

In the Arctic, conventional observations are limited by harsh environmental conditions and logistical challenges. Satellite remote sensing, therefore, plays a crucial role in monitoring sea ice conditions. Infrared (IR) sensors are commonly employed for retrieving IST, as they measure thermal emissions from the uppermost micrometers of snow-covered sea ice (Kang et al., 2023). However, IR-based retrievals are highly susceptible to atmospheric contamination from clouds and hydrometers, resulting in frequent data gaps and spatiotemporal discontinuities (Nielsen-Englyst et al., 2023). Consequently, operational

models often struggle to obtain physically consistent IST data, and also despite their recognized scientific relevance, quantitative assessments of the impact of accurate IST initialization on forecast accuracy remain limited.

To address this limitation, we employ a standalone sea ice model that is periodically nudged with satellite-retrieved ISTs (Kang et al., 2021) to produce a spatiotemporally continuous, gap-free IST dataset. This dataset—hereafter referred to as the "satellite-based IST derived from the sea ice model"—maintains physical consistency with observational constraints while also adhering

to the governing equations of the sea ice model (Guemas et al., 2014). We then incorporate this IST dataset into the Korean Integrated Model (KIM), a global NWP system operated by the Korea Meteorological Administration (KMA). KIM is well suited for evaluating the impact of realistic IST initialization, as it continues to uniformly initialize IST at 271.35 K (−1.8 °C) over all sea ice-covered areas, irrespective of seasons and prevailing atmospheric conditions. Moreover, while KIM performs competitively in mid- and low-latitude regions, its forecasting skill in the Arctic, especially during winter, remains limited

(Choi et al., 2024), despite ongoing updates to its physical parameterizations and data assimilation (DA) techniques.

KIM's simplified assumption ignores Arctic winter surface conditions, where prolonged radiative cooling under polar night can drives near-surface air temperatures down to −40 °C under clear skies (Batrak and Müller, 2019). Consequently, the



operational IST in KIM is typically 10–20 K warmer than actual Arctic winter conditions. This discrepancy likely induce excessive upward heat fluxes from the sea ice surface during the early forecast lead times, potentially leading to a warm bias in the lower troposphere. Such biases could may weaken the surface-based temperature inversion and, in turn, disrupt the stable boundary layer structure. Therefore, this inadequacy in the treatment of sea ice surface conditions can be considered one of key factors limiting the forecasting skill of KIM in Arctic winter.

In this study, we evaluate the forecast impact of initializing realistic IST using observationally constrained dataset described above, comparing it against forecasts initialized with the default operational IST configuration. Although our investigation is conducted within the KIM framework—which currently initializes IST using a spatially uniform and unrealistically warm temperature field—the insights gained are broadly applicable to other NWP models, many of which face similar challenges regarding IST initialization. Our aim is to demonstrate the potential benefits of using physically realistic IST initialization to enhance forecast accuracy in high-latitude regions and provide guidance for future improvements to KIM and analogous forecasting systems The remainder of this paper is structured as follows: Sections 2 and 3 provide an overview of the KIM framework and datasets used; Section 4 describes the experimental design; Section 5 presents the results and associated analyses; and finally, Section 6 summarizes the main conclusions and discusses directions for future study.

## 2 KIM

The KIM is a global, non-hydrostatic NWP model developed by the Korea Institute of Atmospheric Prediction Systems (KIAPS). KIM uses a spectral-element dynamic core on a cubed-sphere grid and a comprehensive physics package comparable to other NWP models. It is coupled with a hybrid four-dimensional ensemble-variational (Hybrid 4D-EnVar) DA system alongside a land DA system. Detailed descriptions of the KIM dynamics, physics, and DA methodologies can be found in Hong et al. (2018), Kwon et al. (2018), Kang et al. (2018), and references therein.

In its atmospheric DA process, KIM assimilates a wide range of observations (e.g., conventional data and satellite radiances) within an ±3 h assimilation window (Shin et al., 2022). To minimize noise resulting from imbalances in the initial state, KIM adopts the incremental analysis update (IAU) method (Bloom et al., 1996) and distributes observation-derived innovations across the assimilation window. This technique helps mitigate spurious dynamic adjustments and ensures a more balanced initialization. Surface analyses within KIM are performed by updating the background states with externally supplied datasets at predefined intervals. Specifically, oceanic boundary conditions, such as sea surface temperature (SST) and sea ice concentration (SIC), are updated in batch at the beginning of the assimilation window (i.e., −3 hours from the analysis time) using the Level 4 Operational Sea Surface Temperature and Sea Ice Analysis (OSTIA) dataset (Shin et al., 2022). OSTIA is a global, daily composite dataset created through optimal interpolation of SST and SIC products derived from multiple sources, including in situ measurements as well as observations from ships, aircrafts, and satellites (Donlon et al., 2012). Notably, OSTIA assigns a fixed temperature of 271.35 K (−1.8 °C) to regions where SIC ≥ 50%, representing the freezing point at



typical ocean salinity (Rayner et al., 2003). Importantly, this temperature corresponds to the sea ice–ocean interface temperature at the base of the sea ice rather than the skin temperature at its top.

This study uses KIM version 3.7 for both deterministic and ensemble forecasts. The deterministic configuration has a horizontal resolution of 25 km and 91 vertical pressure levels extending up to 0.01 hPa. The ensemble system comprises 50 members, each with the same vertical resolution but a coarser horizontal resolution of 50 km. KIM classifies grids as sea ice or ocean based on a 50% SIC threshold; grids with SIC ≥ 50% are treated as sea ice and those with SIC < 50% as ocean. In KIM's surface scheme, sea ice grid is represented by 1D system consisting of one snow layer and two sea ice layers. Snow depth and sea ice thickness are climatologically prescribed. Currently, KIM operationally updates only the IST among sea ice thermodynamic variables without directly constraining other components of surface energy budget or internal layer temperature profile. As such, only the skin temperature at the air–snow interface is initialized from the OSTIA, and other subsurface layer temperatures are inherited from the background state The DA cycles in KIM occur every 6 hours (00, 06, 12, and 18 UTC), bridged by a 9-h forecast. Additionally, 5-d forecasts are lunched twice daily (00 and 12 UTC). A post-processing module that remaps the model outputs to align with ERA5 resolution is also included in order to facilitate direct comparisons.

## 3 Used Data

In this study, a satellite-based, physically realistic IST is generated using a standalone sea ice model and incorporated into the OSTIA dataset, which currently serves as the input for the KIM system. This approach allows the satellite-constrained IST to be seamlessly integrated into the KIM initialization process, ensuring its effective use while maintaining consistency with the current DA configuration. Then, numerical experiments through a modified OSTIA dataset are conducted to evaluate the impact of this revised IST initialization on atmospheric temperature forecasts. The experimental period spans from December 15, 2021 to January 24, 2022, with the first 11 days allocated for model spin-up. The following subsections describe the generation of satellite-based IST dataset, along with the ERA5 data used as reference for validation.

### 3.1 Satellite-Based IST Derived from Sea Ice Model

We simulate IST over the Arctic Ocean using a standalone sea ice model that integrates an 1D thermodynamic framework with nudging and 2D Lagrangian tracking techniques. This model targets regions with SIC > 90%, treating them as fully ice-covered (SIC = 100%). It features a vertical column of snow and sea ice in multiple layers and resolves energy exchanges at the air–snow and sea ice–ocean interfaces, as well as internal heat conduction. The sea ice bottom temperature is fixed at −1.8 ℃ (the seawater freezing point), while its upper boundary is forced from reanalysis-based atmospheric conditions, with snowfall accumulating as an additional layer on top of the existing snowpack.

In the sea ice model, to enhance the accuracy of the IST simulations, a nudging term is applied at the uppermost snow layer, gently steering the model solution toward observational constraints. The nudged data is the high-latitude Level 2 IST product,



provided by Ocean and Sea Ice Satellite Application Facility (OSI SAF). This satellite-retrieved IST product is generated using
multiple regression algorithm based on a split-window technique that leverages the thermal IR channels (10.8 and 12.0 µm)
from the Advanced Very High-Resolution Radiometer/3 (AVHRR/3) onboard the Meteorological Operational (MetOp)
satellite (Dybkjær et al., 2018; Dybkjær and Eastwood, 2018). To enable a 4D representation of sea ice evolution, the 1D
vertical column is extended into a 2D horizontal domain over time by using ice motion vectors provided by OSI SAF. This
satellite-derived motion product is generated using an optimized pattern-matching algorithm that tracks sequential images from
passive microwave radiometers and active microwave scatterometers (Lavergne et al., 2010). Validation of model-simulated
ISTs against drifting buoy temperature measurements indicates a mean deviation of less than 1 K and a root-mean-square
deviation of approximately 2 K. The IST fields are produced every 3 hours at a 25 km resolution. Further details of the
standalone sea ice model can be found in Kang et al. (2021) and references therein.

To efficiently integrate the realistic IST into KIM, while maintaining compatibility with its current DA system, the OSTIA
IST is replaced with the satellite-based IST derived from the sea ice model. Given the daily resolution of OSTIA, the 3-hourly
IST is temporally averaged to daily values. This is justified by the minimal diurnal temperature variation in the Arctic winter
under polar night conditions. The daily-averaged IST is then spatially collocated with the OSTIA grid and further extrapolated
across marginal ice zones to ensure full spatial coverage. Finally, the resulting satellite-based IST($IST_{sat}$) is blended with
271.35 K—representing the temperature of open ocean coexisting with sea ice—using the OSTIA SIC ($0 \leq SIC \leq 1$), as follows:

$$IST_{new} = IST_{sat} \times SIC + 271.35 \times (1 - SIC) \qquad (1)$$

where $IST_{new}$ represents the final blended IST used to replace OSTIA IST values entirely during the experimental period.
Figure 1 presents the spatial distributions of the 41-d mean temperatures over high-latitude regions in the Northern Hemisphere
from both the original and modified datasets. The satellite-constrained dataset shows no noticeable temperature discontinuities
across either the central Arctic Ocean or the marginal sea ice zones, indicating the successful generation and integration of the
revised IST. The blending procedure smooths the temperature transition between ocean and sea ice surfaces, thereby effectively
avoiding artifacts. Moreover, as shown in Fig. 2, marginal sea ice zones are confined to narrow bands along the sea ice edge
throughout the experimental period, implying that their influence on the overall experimental results is limited and can be
considered negligible.

## 3.2 ERA5 Reanalysis

ERA5 is a fifth-generation global reanalysis dataset produced by the European Centre for Medium-Range Weather Forecasts
(ECMWF). Built upon the Integrated Forecast System (IFS) Cy41r2, it incorporates a hybrid 4D-EnVar DA system and a land
DA system. This study utilizes ERA5 atmospheric profiles as reference for validation, provided at 6-h intervals (00, 06, 12,



and 18 UTC) on 31 vertical pressure levels (1000 to 1 hPa) with a horizontal resolution of 0.25°. Detailed descriptions of ERA5 are available in Hersbach et al. (2020), Bell et al. (2021), and related literature.

## 4 Experimental Design

We hypothesize that the current IST initialization in KIM, which uses unrealistically warm values, contributes to a systematic warm bias in the lower troposphere over high-latitude regions at the analysis time. To quantify this impact, we perform a simplified energy budget analysis using a conceptual two-layer atmospheric model. Two IST scenarios are considered: the operational value of 271.35 K (unrealistic warm) and a more physically realistic value of 258.15 K. All other parameters are derived from the January climatology of ERA5 reanalysis for polar regions. The analysis result suggests that the IST difference between the two scenarios would result in approximately a 0.4 K deviation in the lower tropospheric temperature (see Appendix A for details). Thus, if there is a warm bias at the analysis time, correcting the unrealistically high IST to a physically realistic value is expected to reduce the bias by roughly 0.4 K and consequently improve forecast accuracy.

To isolate the impacts of revised IST initialization, we conducted two numerical forecast experiments. The control run (CTL) utilizes the original OSTIA dataset with the standard KIM configuration, whereas the experimental run (EXP) employs the modified OSTIA dataset for initialization. The experiments consist of 120 DA cycles at 00, 06, 12, and 18 UTC from December 26, 2021, to January 24, 2022 (a total of 30 days), including 5-d forecasts initiated twice daily at 00 and 12 UTC. Thus, differences in the analyses and subsequent forecasts between CTL and EXP are attributed solely to revised initial sea ice conditions, potentially revealing indirect and cumulative effects on atmospheric state.

Prior to conducting the forecast experiments, we verify the successful integration of the revised OSTIA dataset into KIM's DA system. Figure 3 exemplifies the skin temperature distributions from both CTL and EXP at the time when the OSTIA dataset is batch-replaced in the background state, 3 hours before the analysis time (e.g., December 15, 2021 at 03 UTC). In CTL (Fig. 3a), unrealistic temperature discontinuities and deviations of approximately 10–20 K are found in the regions between the sea ice and land as well as between the sea ice and ocean. Such abrupt transitions potentially degrade model stability and forecast reliability. In contrast, EXP (Fig. 3b) displays a smoother, physically consistent temperature distribution, confirming that the modified OSTIA dataset successfully mitigates the artificial discontinuities seen in CTL.

## 5 Evaluation Results

In both CTL and EXP, the analysis and forecast fields are evaluated against the ERA5 reanalysis to quantify performance changes, examining systematic errors (hereafter bias) and random errors calculated regionally and across various forecast lead times. Particular focus is given to tropospheric temperature rather than water vapor due to the extremely cold and dry Arctic winter conditions (Fig. S1). Results are detailed in subsequent subsections.



### 5.1 Analysis-Time Impact

*a. Vertical Thermal Structure*

We first evaluate how IST modifications affected the vertical temperature structure of the troposphere at the analysis time.
Bias and random error in tropospheric temperatures are examined from December 26, 2021 to January 24, 2022. As KIM
updates IST in batches 3 hours before the analysis time, the state at the analysis time reflects the subsequent three-hour
numerical integration. Figure 4 shows the time series of lower-tropospheric temperature biases and random errors over high-
latitude regions (above 70° N). EXP exhibits an intensified cold bias between 1000 and 900 hPa compared to CTL, persisting
throughout the entire experimental period. Specifically, mean biases in EXP increase in magnitude from −0.18 K to −0.85 K
at 1000 hPa, from −0.24 K to −0.60 K at 950 hPa, and from −0.30 K to −0.43 K at 900 hPa. The difference in bias between
EXP and CTL gradually diminishes with altitude and becomes negligible above 850 hPa (not shown), indicating that the impact
of IST modifications is largely confined to the shallow lower troposphere, within the lowest 1–2 km of the atmosphere.

Contrary to initial expectation, CTL exhibits a slight cold bias rather than the anticipated warm bias despite unrealistically
warm IST initialization. The revised IST initialization further cools the lower troposphere, intensifying the existing cold bias
in the CTL. Nonetheless, the absolute magnitude of this shift (~0.4 K) aligns well with our conceptual energy budget analysis.
Meanwhile, random errors are modestly reduced in EXP: from 2.54 K to 2.41 K at 1000 hPa (−5%), from 2.11 K to 2.00 K at
950 hPa (−5%), and from 1.68 K to 1.64 K at 900 hPa (−3%). These reductions, while notable, are less substantial than the
increase in bias.

Figures 5a–c present zonal-mean cross-sections of temperature biases across the Northern Hemisphere troposphere (100–1000
hPa) for EXP, CTL, and their differences, respectively. Figures 5d–f show the corresponding random errors. Bias changes are
clearer than changes in random errors, especially between 70–90° N (Fig. 5c vs. 5f). Notably, in the 83–90° N band, CTL
shows a warm bias in the lower troposphere, which is mostly eliminated in EXP (blue box in Figs. 5a–b). In contrast, in the
70–83° N band, an existing cold bias in CTL becomes more intensified in EXP (red box in Figs. 5a–b). Given that the 70–83°
N band covers a broader geographical area than the 83–90° N band, the net effect results in an amplification of the cold bias.
Interestingly, within the 70–83° N band, random error decreases concurrently with the increased bias, implying a possible
trade-off between bias and random error.

*b. Horizontal Thermal Distribution*

Figure 6 displays the spatial distributions of bias and random error differences between EXP and CTL at 1000 hPa, where
changes are most prominent. The corresponding spatial distributions of the biases and random errors individually for EXP and
CTL are shown in Fig. S2, wherein CTL exhibits warm biases in the central and western Arctic Ocean and cold bias in the
eastern Arctic Ocean. The revised IST initialization in EXP results in substantial near-surface cooling confined to sea ice-



covered regions, clearly attributable to modified sea ice initialization (Fig. 6a). Land regions, unaffected by IST modifications, showed negligible differences.

Spatial differences in bias are distinct (Fig. 6b): regions with multi-year sea ice (central and western Arctic Ocean) show
reductions in warm biases, whereas regions with first-year sea ice (eastern Arctic Ocean) experience intensified cold biases. This bias pattern may be suspected to stem from the use of climatological snow depth and sea ice thickness, which does not capture their recent variability. While the zonal-mean cross-section indicates a reduction in random error for EXP within the 70–83°N band, the corresponding spatial distribution (Fig. 6c) reveals no consistent or spatially coherent pattern in the random error differences between EXP and CTL.

**5.2 Forecast-Time Impact**

*a. Vertical Thermal Structure*

To assess the temporal evolution of the tropospheric temperature errors, we examine the mean differences in absolute biases between EXP and CTL over the polar region (north of 70° N) across forecast lead times up to 120 hours (Fig. 7a). Negative values indicate improved performance in EXP relative to CTL, whereas positive values represent degradation. At the analysis
time, positive values of 0.1–0.7 K are shown between 900–1000 hPa, indicating the intensified cold bias in EXP compared to CTL. Although the magnitude of this degradation gradually decreases with forecast lead time, it persists throughout the 5-d forecast period. Figure 7b displays the corresponding differences in the random errors. At the analysis time, EXP achieves a 3–5% reduction in random errors within the lower troposphere; however, this improvement weakens progressively and eventually transitions into a slight degradation throughout the troposphere at the 120-h forecast lead time.

Figure 8 presents the zonal-mean cross-sections of temperature biases and random errors for EXP and CTL at the 120-h forecast lead time, alongside their differences. Both experiments show a persistent cooling tendencies in the lower troposphere over the polar regions (Figs. 8a–b). Although EXP remains cooler than CTL, the magnitude of the bias difference is considerably smaller than that observed at the analysis time (Fig. 5c vs. Fig. 8c). Regarding random error, the difference between EXP and CTL shifts from an initial improvement to eventual degradation (Fig. 5f vs. Fig. 8f), expanding vertically from the lower to
the upper troposphere over the Arctic. Similarly, the random error itself—initially concentrated in the lower troposphere at mid- to high latitudes—progressively propagates upward into the upper troposphere as the forecast advances (Figs. 5d–e vs. Figs. 8d–e).

Importantly, differences in both bias and random error between EXP and CTL remain relatively minor in the low- and mid-latitudes compared to that in high-latitude region, even after forecast day 5. This suggests that the revised IST initialization
does not significantly disrupt large-scale atmospheric circulation within the short-range forecast timeframe. Consequently, the

limited influence of Arctic surface boundary conditions on mid-latitude forecast skill of interest may explain why the deficiency in KIM's IST initialization has remained unrecognized for so long.

*b. Horizontal Thermal Distribution*

Figure 9 illustrates the spatial distributions of bias and random error differences between EXP and CTL at 1000 hPa for the 120-h forecast lead time. Corresponding bias and random error distributions for each experiment are shown in Fig. S3. Over the Barents Sea, EXP induces substantial cooling, intensifying cold bias (Figs. 9a–b). Given that this region frequently experiences intrusions of warm and moist Atlantic air, such a change may subtly but persistently influence the jet stream positioning and associated storm tracks. Random error differences (Fig. 9c), however, exhibit no distinct regional features. Notably, by the 120-h forecast lead time, the magnitudes of both bias and random error differences between EXP and CTL become comparable, in contrast to analysis time when bias differences dominated. This indicates that while bias differences remain relatively steady over time, the random error becomes increasingly perturbed with forecast lead time, eventually reaching a magnitude comparable to that of the bias difference by around forecast day 5.

**6 Conclusions and Discussion**

As Arctic amplification accelerates and sea ice conditions deviate further from historical averages, NWP models should evolve to better represent the polar environment. This study has demonstrated that although accurate IST initialization is critical for improving Arctic forecasts, its independent improvement yields limited benefit unless accompanied by corrections to internal model deficiencies. Specifically, the control run (CTL) in KIM, despite being initialized with an unrealistically warm IST, exhibited the slight cold bias in the lower troposphere, indicating an inherent cooling tendency in KIM. Unexpectedly, the experimental run (EXP), which employed a satellite-constrained, physically realistic IST, further intensified this cold bias. Although EXP reduced random error modestly, the amplified bias ultimately degraded overall forecast performance.

This seemingly paradoxical results underscore the importance of considering model-specific physics when interpreting the effects of surface-driven adjustments. Internal evaluations by KIAPS have identified systematic differences in the cloud microphysics scheme between KIM and ECMWF IFS. Specifically, KIM tends to overestimate ice-phase hydrometeors while underrepresenting liquid water content in polar clouds. Given that liquid clouds trap outgoing longwave radiation more effectively than ice clouds, this imbalance likely weakens cloud radiative feedback, resulting in enhanced radiative cooling in the lower troposphere. As a result, this cooling tendency acts as a compensatory mechanism, counterbalancing the artificial heating introduced by unrealistic warm surface boundary conditions. Thus, a physically realistic IST initialization unintentionally exacerbate underlying model deficiency. Moreover, such boundary initialization issues can reduce the impact of DA for surface-sensitive lower tropospheric observations (Lawrences et al., 2019; Kang et al., 2025), thereby hindering appropriate technical applications.



Recognizing these circumstances, KIM development team has acknowledged deficiency in the current cloud physics scheme and is planning targeted improvements. While this independent update may help mitigate excessive tropospheric radiative cooling, it could also introduce new warm biases driven by unrealistic surface boundary conditions. Therefore, integrating the satellite-based IST initialization approach proposed in this study with physics upgrades may offer a more balanced and effective solution, ultimately contributing to the reduction of forecast errors. Future work will focus on investigating the interaction between IST initialization and improved cloud physics, with the goal of supporting broader efforts to enhance polar forecast skill in global NWP systems.

**Appendix A.**

A simplified energy budget analysis was conducted using a two-layer conceptual model to quantify the impact of IST modification on the lower-tropospheric temperature. This model, which is illustrated in Fig. A1, accounts for the radiative energy exchange between atmospheric layers, along with the surface energy balance. Two IST scenarios were considered: Scenario 1 ($T_s$ = 271.35 K), which represented the default initialization in the KIM, and Scenario 2 ($T_s$ = 258.15 K), which considered a more physically realistic value. The objective of this analysis was to estimate the temperature adjustments in the lower troposphere induced by variations in IST initialization.

The system consisted of three main components: 1) The upper atmosphere (U), which extended from 850 hPa to the top of the atmosphere and had temperature $T_U$ = 235 K and radiative absorption coefficient $a_U$ = 0.4; 2) the lower atmosphere (L), which extended from the surface to 850 hPa and had temperature $T_L$ = 248.15 K and radiative absorption coefficient $a_L$ = 0.1; 3) the surface boundary where the two IST scenarios were applied, which had a radiative absorption coefficient of 1.

The upper atmosphere emits downward radiation at a rate of $a_U\sigma T_U^4$, where $\sigma$ is the Stefan–Boltzmann constant ($5.670 \times 10^{-8}$ W m$^{-2}$ K$^{-4}$). The lower atmosphere absorbs a fraction of this radiation, given by $a_L a_U \sigma T_U^4$, and emits radiation both upward and downward at a rate of $a_L\sigma T_L^4$. Similarly, the surface emits upward radiation at a rate of $\sigma T_S^4$, with the lower atmosphere absorbing $a_L\sigma T_S^4$. Under Arctic winter conditions, the latent heat flux is neglected, whereas the sensible heat flux ($F_S$) is the dominant mechanism for heat exchange between the surface and atmosphere. The sensible heat flux is calculated as follows:

$$F_s = -\rho_{air} c_{p,air} C_s U_{ns} (T_{ns} - T_s) , \tag{A1}$$

where $\rho_{air}$ = 1.41 kg m$^{-3}$ is the air density, $C_{p,air}$ = 1005.0 J kg$^{-1}$ K$^{-1}$ is the specific heat capacity of air, $C_s$ = 0.001397 is the bulk transfer coefficient for sensible heat, $U_{ns}$ = 5 m s$^{-1}$ is the near-surface wind speed, and $T_{ns}$ = 257.15 K is the near-surface air temperature.



To estimate the net energy change in the lower atmosphere during 3 hours, constant heat fluxes are assumed, and the energy balance equation is defined as follows:

$$Q_{net} = Energy\ Gain - Energy\ Loss\ ,$$
(A2)

where the energy gain is denoted by

$$Energy\ Gain = a_L a_U \sigma T_U^4 + a_L \sigma T_s^4 + F_S\ ,$$
(A3)

and the energy loss is calculated using

$$Energy\ Loss = 2a_L \sigma T_L^4\ .$$
(A4)

The heating rate (HR, in K) of the lower atmosphere is then determined as follows:

$$HR = \frac{Q_{net}}{\rho_{air} C_{p,air} h} \cdot t\ ,$$
(A5)

where h = 1224 m is the height of the lower atmosphere and t = 10800 s (3 h) is the duration of the analysis. The initial $F_S$ was the highest in Scenario 1 ($T_s$ = 271.35 K; $T_{ns}$ = 257.15 K). However, as the environment rapidly adjusted, the skin temperature cooled rapidly, reducing the temperature difference ($T_{ns} - T_s$) obtained from Eq. A1 over time and eventually converging
toward the value seen in Scenario 2 by the end of the 3-h period. Considering the anticipated rapid skin temperature adjustment, the temperature difference was assumed to decrease linearly during 3 hours; thus, half of the calculated $F_S$ value was applied for Scenario 1.

The results indicated that in Scenario 1 with an IST of 271.35 K, $Q_{net}$ was 64.93 W m$^{-2}$, resulting in an HR value of 0.394 K over 3 h. Conversely, in Scenario 2 with an IST of 258.15 K, $Q_{net}$ was −1.00 W m$^{-2}$, leading to an HR value of −0.006 K. From
these values, the estimated temperature adjustment in the lower troposphere between the two scenarios can be determined as follows:

$$\Delta T_L \approx 0.4\ K\ .$$
(A6)

These results indicated that initializing the model with an unrealistically high IST could induce artificial warming (approximately 0.4 K in the lower troposphere within the first few hours of model integration). This artificial warming
suggested that the current IST initialization in the KIM may have contributed to a warm bias in the lower troposphere at the analysis time.



*Code and data availability.* The ERA5 reanalysis dataset is publicly available from the Copernicus Climate Change Service (C3S) Climate Data Store (CDS): https://cds.climate.copernicus.eu/datasets/reanalysis-era5-pressure-levels?tab=overview (Copernicus CDS, 2025). The modified OSTIA dataset and the code used for the conceptual two-layer energy budget model

presented in Appendix A are available for download from the Zenodo repository: https://doi.org/10.5281/zenodo.15313858 (Kang and Sohn, 2025).

*Competing interests.* The contact author has declared that none of the authors has any competing interests

*Acknowledgments.* The authors gratefully acknowledge the Korea Institute of Atmospheric Prediction Systems (KIAPS) for providing the research environment that enabled access to supercomputing facilities, which were indispensable for conducting

the numerical experiments in this study.

*Financial support.* This work was supported by Basic Science Research Program through the National Research Foundation of Korea (NRF) funded by the Ministry of Education (No. RS-2023-00271704), and Postdocs (LAMP) Program through the National Research Foundation of Korea (NRF) funded by the Ministry of Education (No. RS-2023-00210362).

*Author contribution.* **EJK**: Conceptualization, Formal analysis, Investigation, Methodology, Software, Validation,

Visualization, Writing – original draft, Funding acquisition, Supervision; **BJS**: Conceptualization, Methodology, Resources, Writing – review and editing; **WK**: Conceptualization, Methodology, Software, Writing – review and editing; **YCN**, **SL**, **IHK**: Conceptualization, Methodology, Writing – review and editing; **HJS**: Writing – review and editing, Funding acquisition.

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




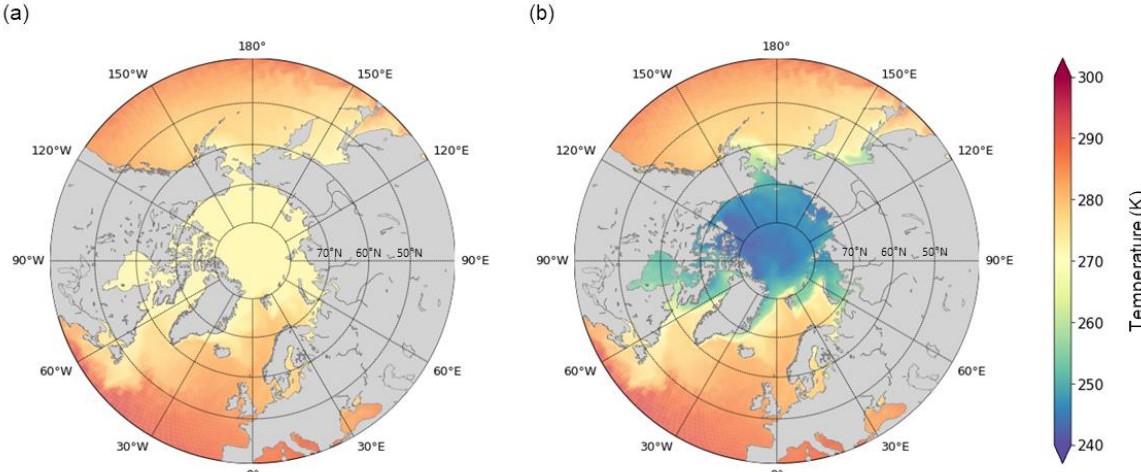

**Figure 1: Spatial distributions of mean temperature from December 15, 2021 to January 24, 2022 based on (a) the original operational sea surface temperature and sea ice analysis (OSTIA) dataset and (b) the new OSTIA dataset over the middle and high latitudes of the Northern Hemisphere.**





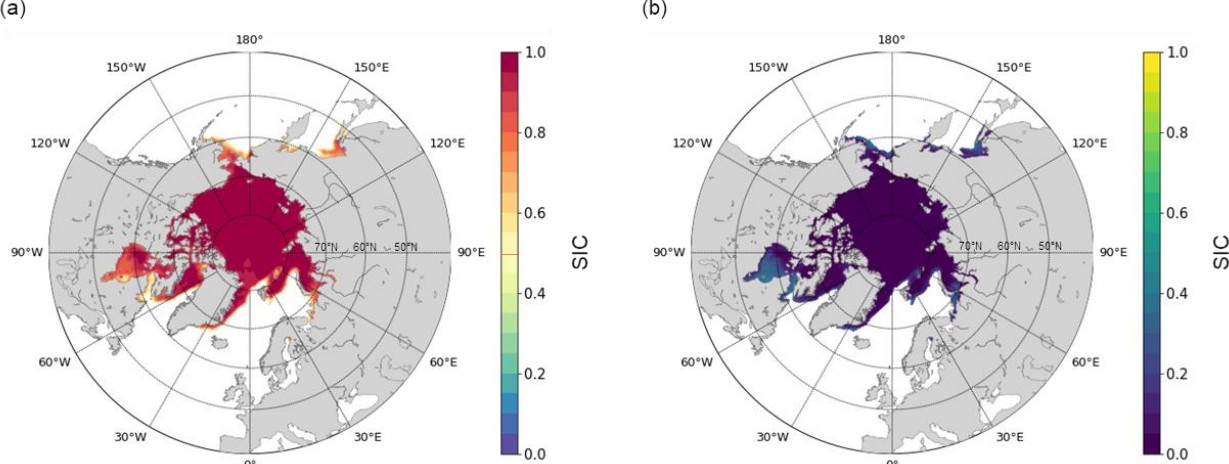

Figure 2: Same as Figure 1, but for sea ice concentration (SIC) in regions where SIC > 0.5. (a) Mean and (b) standard deviation. In the Korean Integrated Model, only regions with SIC > 0.5 are classified as sea ice.



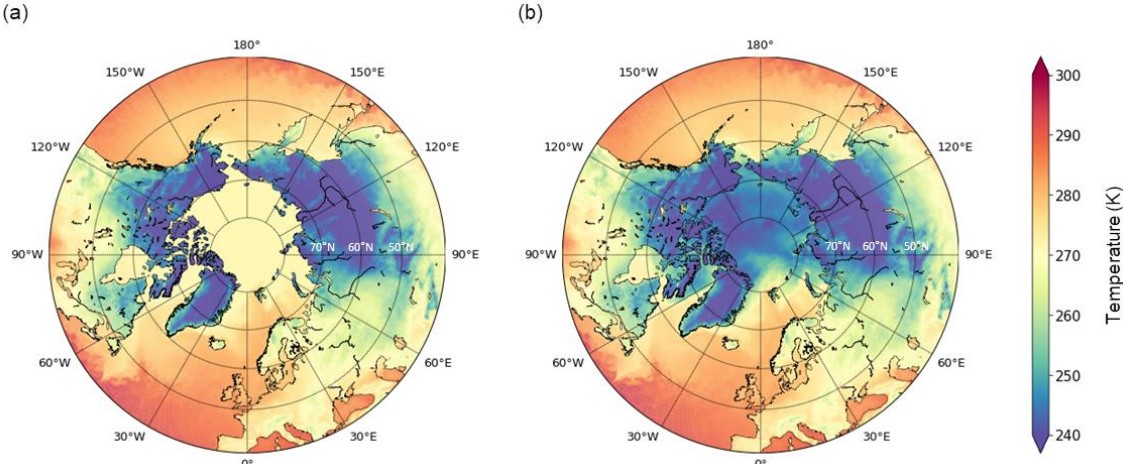

**Figure 3: Spatial distributions of skin temperature after initialization in (a) CTL and (b) EXP at 03 UTC on December 15, 2021.**







**Figure 4: Time series of systematic (bias) and random errors in temperature within the Northern Polar region (>70° N), computed relative to ERA5 reanalysis, from December 26, 2021 to January 24, 2022 at (a) 1000 hPa, (b) 950 hPa, and (c) 900 hPa.**





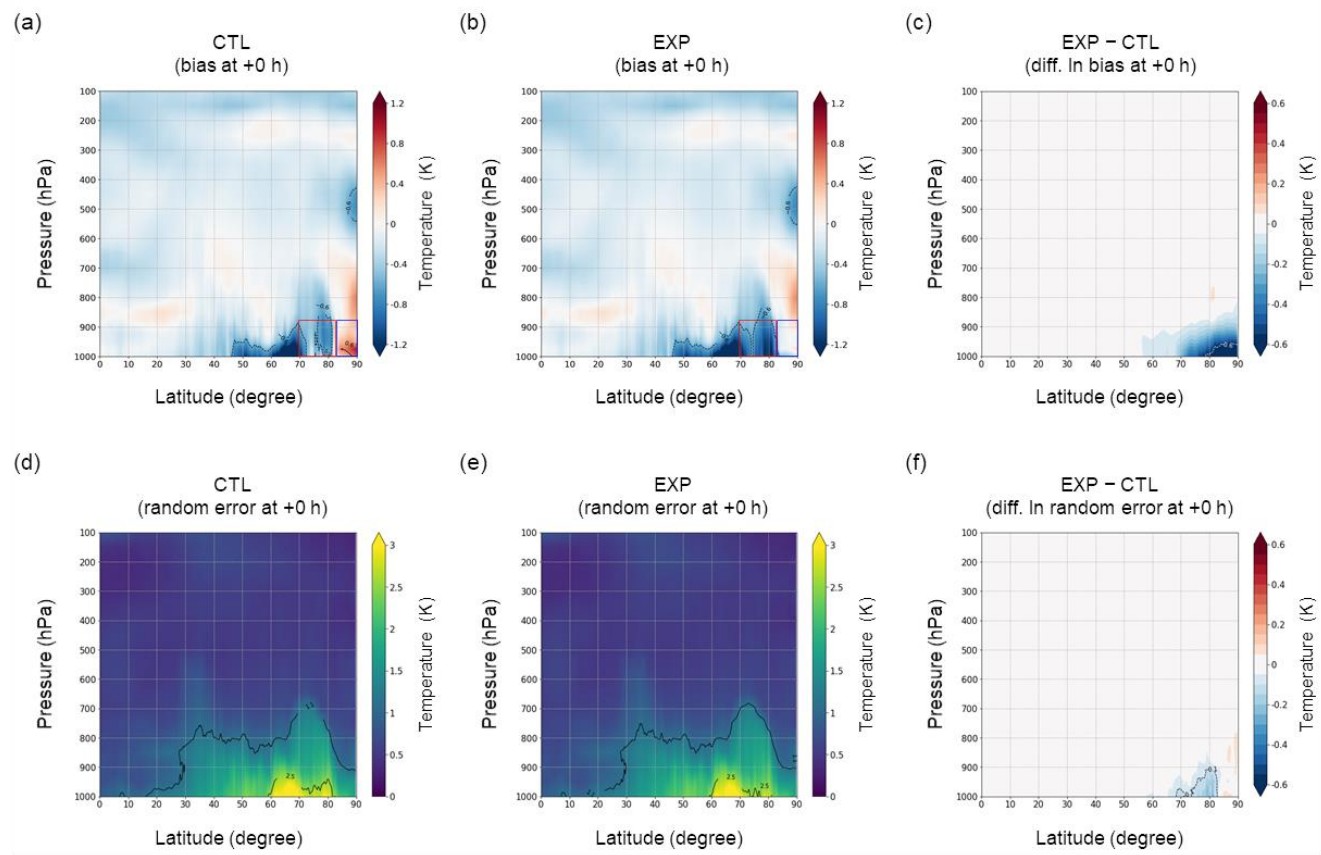

**Figure 5: Zonal cross-sections of bias and random error in Northern Hemisphere tropospheric temperature for CTL and EXP, along with their differences, at analysis time (0 h), averaged from December 26, 2021 to January 24, 2022.**



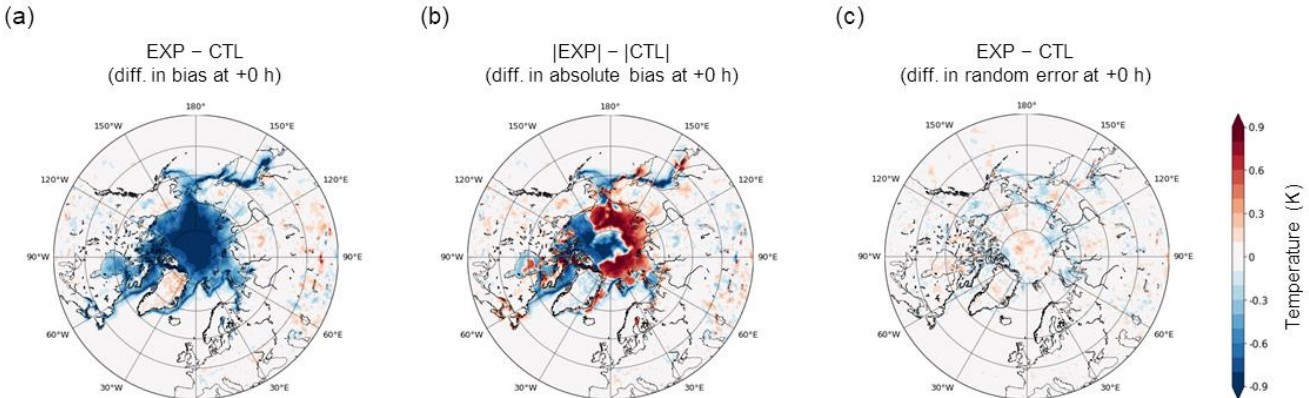

**Figure 6: Spatial distributions of differences in (a) bias, (b) absolute bias, and (c) random error in temperature at 1000 hPa between EXP and CTL at analysis time (+0 h), averaged from December 26, 2021 to January 24, 2022.**




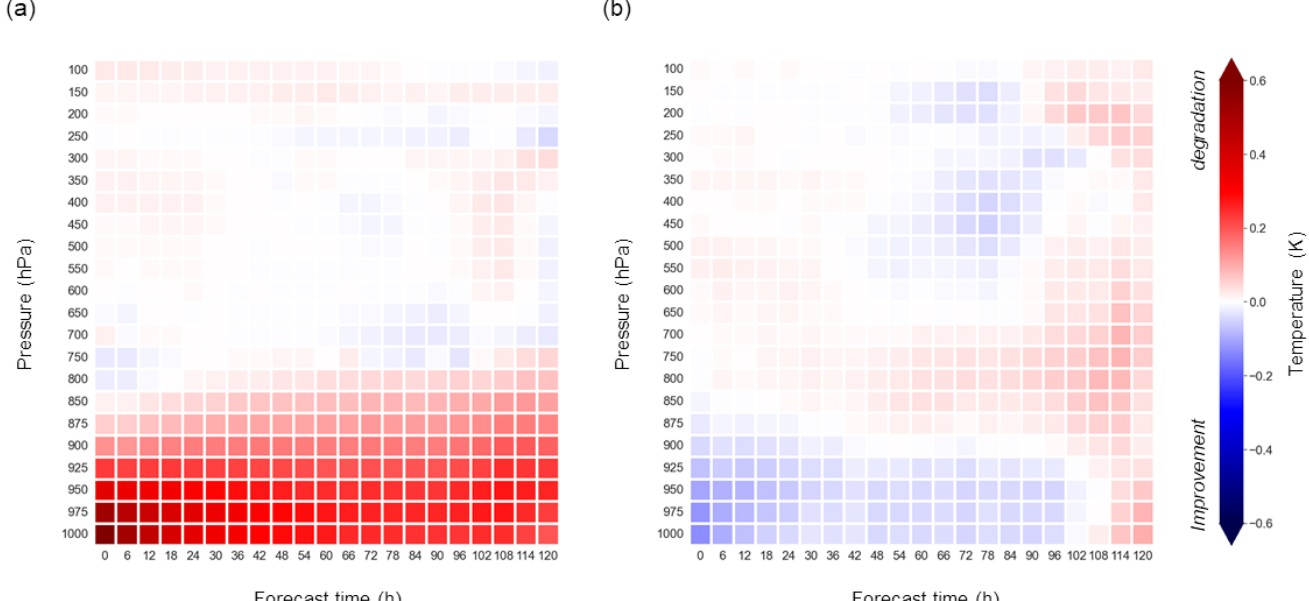

**Figure 7: Temporal evolution of differences in (a) absolute bias and (b) random error in tropospheric temperature between EXP and CTL in the Northern Polar region (>70° N), evaluated at 6-h intervals up to a 120-h forecast lead time, averaged from December 26, 2021 to January 24, 2022. Negative values (blue) indicate improvement, whereas positive values (red) indicate degradation.**



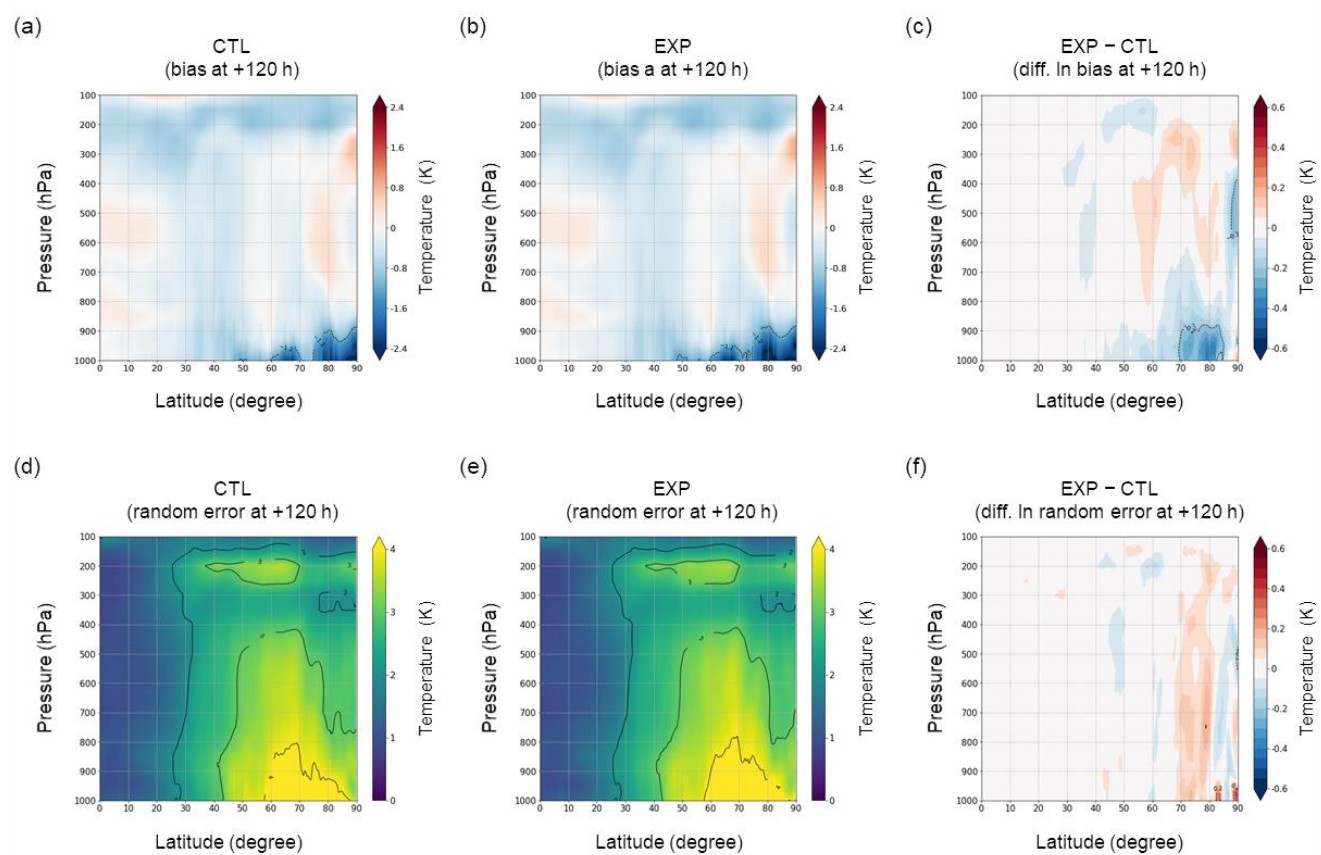

**Figure 8: Same as Figure 5 but for the 120-h forecast lead time (+120 h).**



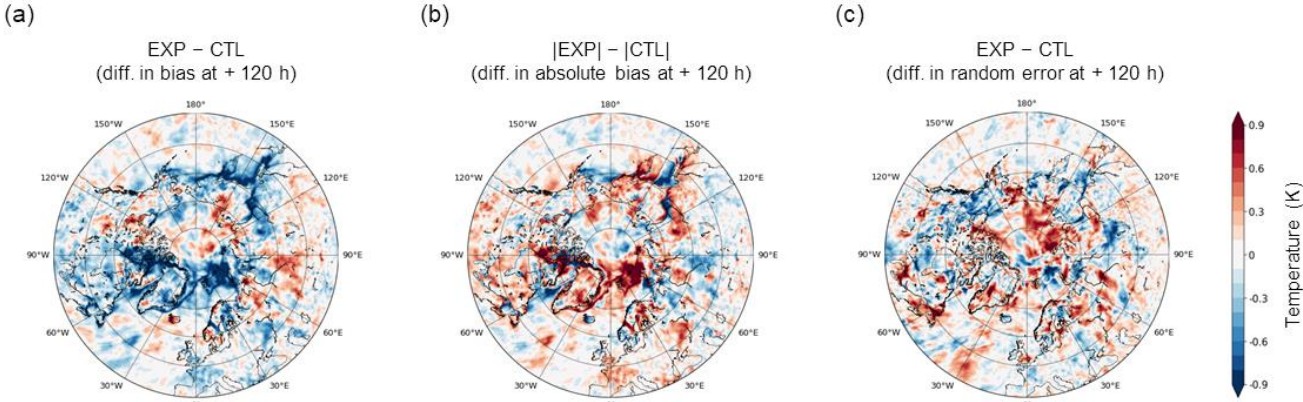

**Figure 9: Same as Figure 6 but for the 120-h forecast lead time (+120 h).**



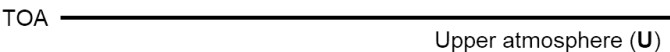

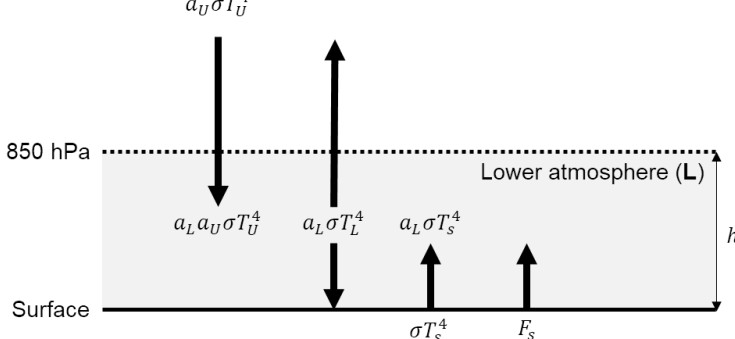

**Figure A1: Schematic representation of the two-layer conceptual model**