# Peer review of "Impact of Satellite-Based Ice Surface Temperature Initialization on Arctic Winter Forecasts Using the Korean Integrated Model"

_EGUsphere, 2025_

## Referee Comment (RC1)

Review of "Impact of Satellite-Based Ice Surface Temperature Initialization on Arctic Winter Forecasts Using the Korean Integrated Model" by Kang et al.

This manuscript highlights the importance of improving the specification of sea ice surface temperature (IST) as a boundary condition in the polar regions for global NWP models, which have traditionally focused on performance in the tropics and mid-latitudes. To this end, the authors conducted an experiment in which the arbitrarily prescribed IST in the KIM model was replaced with a more realistic value to examine how this change affects the polar temperature bias patterns currently seen in KIM. However, the results ultimately indicate that, even with realistic IST, improvements in polar forecast skill are limited unless the model's physical deficiencies are addressed. In this sense, the manuscript implies that better initial conditions alone are insufficient to enhance model performance in the Arctic without improvements to model physics.

Had a realistic IST been applied to an NWP model without physical deficiencies and led to improvement of model bias, it could have provided clear evidence for the importance of the initial boundary condition. However, since a model with known limitations was used and no significant improvement was observed using realistic IST, it is difficult to assess the role of the initial condition in this case. The authors conclude that improvements to the model's physical processes are necessary in order to fully utilize realistic initial conditions. However, this claim is weakly supported by the current analysis and, as such, should be regarded as a diagnosis of the model rather than a confirmed finding. Moreover, while this may be controversial, data assimilation should be regarded as a means to improve model performance—not a justification for altering the model to accommodate more realistic observations. Doing so would, in effect, reverse the proper relationship between the model and the data assimilation process.

Therefore, I encourage the authors to consider revising the manuscript to incorporate the following critical points. As it stands, some of the conclusions are not fully supported by the analysis, and a major revision would be necessary before the manuscript can be considered for publication.

**General Comments:**

**(major points)**

1. The manuscript refers to "realistic IST," but does not provide validation to demonstrate how realistic it actually is. Moreover, the algorithm used (Kang et al., 2021) was only validated using data up to 2015, which is significantly earlier than the experimental period of this study (2022). Considering the observational limitations in polar regions, some form of validation—such as comparisons with IR-based monthly means or other reference datasets—would be necessary to support the accuracy of the realistic IST used in this study.

2. Clarifying the role of microwave (MW) observations in the generation of the OSTIA dataset

would enhance the readers' understanding of the properties of the dataset and is therefore encouraged. Specifically, in the case of the "realistic IST," there appears to be a concern regarding the potential redundancy of MW data, as it is not clear whether MW observations were already assimilated into OSTIA and then additionally used in the current analysis.

3. Including a simple case study showing how the correction of a specific internal model deficiency influences the bias would provide concrete evidence to support the manuscript's claim about the need to improve the internal model processes in KIM.

**(minor points)**

1. Regarding lines 160–163 and Appendix A, it would be important to include further explanation for the assumed low absorption coefficient in the lower atmosphere. Furthermore, if the absorption coefficient were reversed (i.e., 0.1 in the upper atmosphere and 0.4 in the lower atmosphere), Scenario 1 would yield a $Q_{net}$ of 5.9 W m$^{-2}$ (HR value of 0.03), and Scenario 2 a $Q_{net}$ of -54.46 W m$^{-2}$ (HR value of -0.34), leading to a temperature change of approximately 0.37 K. While the values would be similar, this reversed setting appears to better explain the absolute magnitude of the shift described in Section 5.1, particularly when using absorption coefficient of 0.1 in the upper and 0.4 in the lower atmosphere.

2. To better support the discussion in lines 184–192, it would be beneficial to provide an additional figure presenting the vertical profile of the bias, which could aid in clarifying its distinct features.

3. In lines 199–206, it may be advisable to include a figure analogous to Figure 4, focused specifically on the latitudinal range of 83°N–90°N, to more clearly isolate and highlight the localized effect.

4. To enhance the interpretability of Figure 6, it is recommended to include a difference plot in Figure 3. Additionally, presenting the sea ice distribution at the corresponding time (e.g., SIC) would be helpful in explaining the occurrence of unrealistic discontinuities.

**Specific Comments:**

Line 74: "forecasting systems The remainder" → "forecasting systems. The remainder"
Line 91: "global, daily" → "global and daily"
Line 104: "background state The DA cycles" → "background state. The DA cycles"
Figure 2: This figure is not the same as Figure 1. Please rephrase this figure title.
Figure A1: This title is missing a period '.' at the end.

---

## Author Comment (AC1)

Dear Executive and Topical Editors,

Thank you for your feedback regarding our manuscript's compliance with the GMD Code and Data Policy.

We fully recognize the importance of transparency and reproducibility as upheld by GMD, and we acknowledge that the initial submission did not clearly articulate the institutional restrictions related to model code accessibility. We are grateful for the opportunity to correct this oversight and to bring our manuscript into alignment with the journal's policy.

The Korean Integrated Model (KIM) is developed and maintained by the Korea Meteorological Administration (KMA) and, due to institutional and national policy constraints, its source code is not publicly available. These restrictions are beyond our control as authors. This situation is comparable to other operational NWP systems, such as ECMWF's Integrated Forecasting System (IFS), for which GMD has acknowledged similar limitations in recent publications. For example:

> "*Model codes developed at ECMWF are the intellectual property of ECMWF and its member states and are therefore not publicly available*."
> (e.g., https://doi.org/10.5194/gmd-2024-188; https://doi.org/10.5194/gmd-17-7539-2024)

In accordance with the GMD Code and Data Policy clause:

> "*Where the authors cannot, for reasons beyond their control, publicly archive part or all of the code and data associated with a paper, they must clearly state the restrictions.*"

We will explicitly reflect this limitation in the "Code and Data Availability" section of our manuscript, as follows:

"The Korean Integrated Model (KIM), developed and maintained by the Korea Meteorological Administration (KMA), is not publicly available due to institutional and national policy restrictions. Access may be granted to approved partners of KMA in accordance with internal policies."

Furthermore, in line with the policy that:

> "*Where only part of the code or data is subject to these restrictions, the remaining code and/or data must still be publicly archived.*"

We have already made available the key boundary condition datasets and conceptual model code through a public repository, and have provided download link for the relevant reference material. We will further expand the repository during the revision process to include additional model output datasets for each experimental case.

We respectfully request that the manuscript be allowed to proceed through peer review under this framework, given that key data and supporting code—excluding only the KIM source code restricted by institutional and national policy—have been made openly accessible in accordance with GMD's policy.

Sincerely,
Euijong Kang

---

## Author Comment (AC2)

Dear Executive and Topical Editors,

Thank you very much for your kind reply and clarification.

1.  We have re-confirmed the institutional policy by contacting the Korea Meteorological Administration (KMA) regarding the external availability of the KIM source code. KMA has stated that, in accordance with the *National Research and Development Innovation Act*, KIM is currently not subject to public release during the ongoing development phase, and that the decision regarding its future disclosure will be made upon the completion of development.

    The relevant legal basis, the *National Research and Development Innovation Act*, is available only in Korean. For your reference, the official link to the law is provided below:
    https://www.law.go.kr/lsInfoP.do?lsId=013774&ancYnChk=0#0000

2.  KIM has been undergoing an advanced development program since 2020, led by the Korea Institute of Atmospheric Prediction Systems (KIAPS), and officially entered operational service in 2022 as Korea's national NWP system. As the national forecasting infrastructure, the KIM is securely and systematically archived to ensure long-term preservation, version integrity, and sustainable operational maintenance.

We respectfully request your understanding in this regard.

Best regards,

Eui-Jong Kang